# Deterministic Strided and Transposed Convolutions for Point Clouds Operating Directly on the Points

## Abstract

The application of Convolutional Neural Networks (CNNs) to process point cloud data as geometric representations of real objects has gained considerable attention. However, point clouds are less structured than images, which makes it difficult to directly transfer important CNN operations (initially developed for use on images) to point clouds. For instance, the order of a set of points does not contain semantic information. Therefore, ideally, all operations must be invariant to the point order. Inspired by CNN-related operations applied to images, we transfer the concept of strided and transposed convolutions to point cloud CNNs, enabling deterministic network modules to operate directly on points. To this end, we propose a novel strided convolutional layer with an auxiliary loss, which, as we prove theoretically, enforces a uniform distribution of the selected points within the lower feature hierarchy. This loss ensures a learnable and deterministic selection, unlike the iterative Farthest Point Sampling (FPS), which is commonly used in point cloud CNNs. The high flexibility of the proposed operations is evaluated by deploying them in exemplary network architectures and comparing their performances with those of similar (already existing) structures. Notably, we develop a light-weight autoencoder architecture based on our proposed operators, which shows the best generalization performance.

## 1 Introduction

The processing of point clouds is crucial for numerous modern applications. For example, in autonomous driving, LiDAR sensors enable vehicles to create a 3D scan of their surroundings and operate based on the obtained information. While there are several sophisticated CNN architectures for image data, key network modules like convolutions with varying stride and max-pooling cannot be directly applied to point clouds. This is because point clouds are sets of points, which are located arbitrarily in the three-dimensional Euclidean space, and are not structured like pixels in images. Therefore, no grid structure defines how to concatenate the feature vectors of individual points to a tensor that captures all nodes of the point cloud. Consequently, operations on the tensor must be permutation invariant to ensure consistent computation and resemble those of image processing.

While there are several concepts of convolutions with a stride of one that have been applied to point clouds [21, 24, 20, 3], there is no approach that transfers permutation invariant convolutions with a higher stride directly to point clouds without an additional auxiliary grid. Yet for image processing, feature hierarchies enforced with strided convolutions are a crucial concept in many well-known deep learning architectures, e.g., Autoencoders [10], u-net [19], and YOLO [18, 16, 17].

Hence, this work draws motivation from the assumption that transferring deterministic strided and transposed convolutions to point clouds offers great potential. It's main contributions can be summarized as follows:

- We provide a proxy for permutation invariant convolutions with a step size larger than one for point clouds based on an auxiliary loss, which, as we will prove, ensures selection diversity. Complementary, we provide a proxy for transposed convolutions that does not require knowledge of the points. Since both approaches operate directly on the points, they are closely related to the operation of their original counterparts from image-based CNNs.

- Using these building blocks, we construct an autoencoder[1] that not only outperforms the current state-of-the-art for reconstructing the complete point cloud but also generalizes better than existing approaches.

- We show that a properly configured version of our model can learn meaningful high-level features despite being light-weight. Moreover, we demonstrate that our selection strategy can be integrated into existing architectures and replace FPS without a great loss in performance.

The remainder of this paper is organized as follows: The relevant work related to CNNs on point clouds is outlined in Section 2. In Section 3, we introduce our proposed approaches that essentially transfer strided and transposed convolutions from the image domain, placing a particular emphasis on the auxiliary selection loss. Potential applications for the network modules as well as the associated experimental and ablation studies are presented in Section 4. At last, Section 5 concludes this work.

## 2   Related Work on Point Cloud CNNs

One of the first techniques that enabled the use of CNNs and convolution operations for point clouds was to voxelize them. However, these operations cannot be applied directly to point clouds without the supporting grid structure. Moreover, voxelization scales cubically with voxel resolution, so these approaches represent a tradeoff between computational cost and accuracy.

PointNet, introduced by Qi et al. (2017), was the pioneering neural network architecture for applying deep learning directly to point clouds. The concept of the network is to first process each point individually and finally apply global max-pooling to enable the processing of the feature vector with a fully connected neural network. The main disadvantage of PointNet is that it cannot directly incorporate local features of neighboring points into the convolution. The enhanced version PointNet++ [15], was the first network to introduce feature hierarchies while working directly with points. PointNet++ uses iterative farthest point sampling (FPS) [5] to define regions processed by lower-level PointNets, with higher-level features captured in the points sampled by FPS.

The Dynamic Graph CNN (DGCNN) proposed by Wang et al. (2019) introduces two novel ideas: first, EdgeConv, a new type of convolution that operates on the $k$-nearest-neighbor-graph ($k$-NN-graph) of the point cloud, and second, a dynamic update of the $k$-NN-graph giving the network its name. The former is the first convolution operation on points which is conceptually transferred from the image domain. It is based on the observation that in images, the convolutional kernel operates on the eight nearest neighbor pixels (also known as Moore neighborhood) of a pixel. A concept for convolutions with a stride greater than one, however, is missing.

Other approaches to convolutions on point clouds include KPConv [20] and PAConv [24]. Like DGCNN, PAConv processes the $k$-NN-relationships. However, instead of directly learning weights, an assembled weight matrix is predicted. This matrix is used to compute features of neighbor relationships. PAConv can therefore be integrated into existing architectures as a novel and versatile concept. KPConv, contrarily, operates with a spatial kernel instead of relying on the $k$-NN-neighborhood. Notably, KPConv proposes an analogy to convolutions with a stride greater than one, which is based on a grid subsampling strategy with a cell size depending on the radius of the kernels. The points for different hierarchies are determined by the barycenters of the original points in each cell. Hence, the stride approach does not operate directly on the points. Since all information must be incapsulated in a code word, the upsampling approach from KPConv, passing the information gathered in barycenters to the before aggregated points, is not suitable to construct a decoder for this task.

Generally, the approaches for point cloud autoencoders are based on the idea that point clouds describe the surfaces of objects. The approach is to fit a 2D grid by trying to stretch, squeeze and fold it onto the 3D surface. The corresponding origami instructions are saved in the code word of the autoencoder. The first network to propose this idea was FoldingNet [26]. However, this technique

---

[1]The corresponding code can be found in the supplementary material.

struggles when objects possess holes, or multiple objects are present, as the network then has to stretch the 2D grid across empty space. To solve this issue the most recent approach to reconstructing a complete point cloud, TearingNet [13], additionally learns how to cut and tear the 2D grid into the desired shapes. This is done very successfully as it is the state-of-the-art for autoencoding point clouds with multiple objects. Recently, the utilization of transformer architectures has extended to the point cloud domain as well [27, 6, 28]. In particular Point-M2AE [28] leverages FPS-based feature hierarchies and is pre-trained with the self-supervised task of masked autoencoding. Contrarily to complete reconstruction, only parts of the point clouds are covered and reconstructed. They are never seen by the entire model, necessitating the meaningful reconstruction of previously unseen points. Consequently, skip connections between encoder and decoder are permissible as the objective is to uncover points that have not been exposed through those skip connections.

Instead of using stride to reduce the number of processed points as done by image CNNs, other learnable subset approaches have been proposed for point clouds, aiming to improve over FPS with a network-decided and permutation independent selection of points. While according to its definition FPS is permuation invariant, in most applications a greedy or iterative version of FPS is considered to decrease computational complexity. The greedy sampling technique is not permutation invariant as the subsampled output depends on a random starting point and can, thus, result in inconsistent outputs. Gumbel Subset Sampling (GSS) proposed in Yang et al. [25] employs a Gumbel-Softmax to enable a soft learnable selection of points during training and performs a reparameterization during inference. This leads to a Gumbel-Max which then selects specific points in the point cloud. Nevertheless, GSS does not ensure diversely selected points and the final network setup requires a combination of FPS and GSS in order to improve the performance. Critical point layers proposed by Nezhadarya et al. [12] select points based on the number of highest feature activations per point. This operation is permutation invariant. However, an equal distribution of the points is not enforced and it cannot be ensured that the network identifies critical points at the beginning of the training. Finally, Lin et al. [11] present different sampling strategies that are tailored in advance to specific tasks and can be learned by the network. These strategies are designed to enable the network to perform well on their respective tasks, but lack generality.

## 3 Developing Strided and Transposed Convolutions for Point Clouds

As outlined above, there exists a research gap regarding the important concept of convolutions with a stride greater than one operating directly on the points. Currently, iterative FPS is used for this purpose. However, it is not permutation invariant, and hence results in inconsistent outputs. Our approach to convolutions with a stride greater than one operates on the $k$-NN-graph and samples points by itself for higher feature hierarchies. This is done with an auxiliary loss which enforces a uniform distribution of the selected points. Further, we propose a counteracting transposed convolution. The individual network components are described in the following.

**Strided Convolutions**  Generally, in the convolution operation on images, the learnable kernel weights are multiplied by the pixels that match the weights in their position relative to a central pixel. This central pixel has neighboring central pixels that the kernel is applied to as well. The distance between those positions is determined by the stride as its value corresponds to the step size between two kernel positions. For point clouds, the convolutional layers with a stride of one proposed in this paper, operate similarly to those from DGCNN with the nearest neighbors being determined based on the dynamic graph. Slightly deviating from DGCNN, the feature vectors from the current node to its $k$-nearest-neighbors are gathered behind the feature vector of the current node, and a $(1 \times 1)$ convolutional kernel is applied to this tensor. Thus, all neighboring points contribute to the feature result and not just those with the highest activation.

Unlike basic convolution, convolutional layers with a stride greater than one decrease the size of the input, which requires the network to compress the information. While basic convolutions can be used on the $k$-NN-graph, this approach does not work for strides greater than one because there is no inherent method to select the central points. In the image domain, a typical convolution that reduces the size of a feature map has a stride of two and a convolutional kernel of size $(3 \times 3)$. To mimic this convolution in the case of point clouds, the kernel should only be applied on $\lfloor \frac{n}{4} \rfloor$ nodes. To ensure a similar spacing as between the central pixels of images, the points should not overlap each other within the 4-nearest-neighbor neighborhood. However, splitting the point cloud into such subgroups

may not always be possible (e.g., selecting nine points of a uniform $6 \times 6$ point grid) and may not always yield the same result (consider points with equal distances to its 2-nearest neighbors placed on a circle). The main idea to overcome these obstacles and realize a stride greater than one for point clouds, nevertheless, is to let the network decide which nodes to process further in the successive layers and simultaneously enforce a diverse selection of points. Letting the network decide which nodes to process further is implemented with an attention map, i.e., a vector with one importance value per node, predicted by the network. This resembles a score function $s(p_i)$ computing a score for each node $p_i$ of the point cloud $P$. From this attention map, the $\lceil \frac{n}{f_d} \rceil$ highest values and the corresponding node indices are selected. Here, $f_d$ is the factor by which the set of points should be decreased. This means that the value of $f_d$ corresponding to the typical strided convolution is 4. The nodes that have not been selected in the multi-dimensional feature map are then dropped.

**Auxiliary Selection Loss** If the network decides itself which nodes to keep and which not, it may happen that only nodes in an arbitrary fraction of the point cloud are selected at the beginning of the training. Then, a diverse selection of points cannot be simultaneously enforced. This does not correspond to the idea of strided convolutions for images, and the network cannot process the full information. Additionally, the selection may be unstable and unpredictable for the network, causing problems in the learning process. Thus, the network needs to learn which nodes are neighbors in the feature map and preferably select non-neighboring nodes to attain evenly distributed points. To guide the prediction of the attention map referred to above, in this direction, an auxiliary loss capturing the diversity of the selected nodes is computed. In order to do so, for every node, the attention values of its $(f_d - 1)$-nearest neighbors are gathered behind the attention value of the corresponding node in the attention map, resulting in a matrix $M$. Ideally, from a selection diversity perspective, the two following conditions are met: the first entry in each row of the selected points equals one[2], and the other values of this row equal zero. Further, in the rows of the non-selected points, the first entry should be zero and one of the other entries one. Table 1 shows an example for an ideal $M$. If both conditions are met, it is ensured that 1) there are no neighboring selected points (all row entries are zero except for the first one), and 2) every non-selected point has a selected neighbor (one of the entries, except for the first one, is not zero). Then, the selected points are evenly spread in the $k$-NN-graph over which the strided convolution will slide. To measure to what extent the desired properties are met by the network, a loss per row and column is computed. The values in every row are summed, and as it should yield one for every point in the ideal setting, the deviation is measured in terms of the squared error. The column-wise loss is computed only for the selected points, in which case the sum of the first column entries should yield $\lceil \frac{n}{f_d} \rceil$, and the sum of the remaining columns should yield zero. The total loss is the sum of all parts multiplied by $\frac{1}{f_d}$ as a weighting factor in the case of different $\frac{1}{f_d}$ throughout the network. An illustration of the whole selection operation can be seen in Figure 3. Assuming that $m_{i,j}$ is the entry in the $i$th row and the $j$th column of $M$, the mathematical equation for the auxiliary loss $\mathcal{L}_S$ is

$$\mathcal{L}_S = \frac{1}{f_d} \cdot \left( \left[ \left( \sum_{i=1}^{\lceil \frac{n}{f_d} \rceil} m_{i,1} \right) - \left\lceil \frac{n}{f_d} \right\rceil \right]^2 + \sum_{j=2}^{f_d} \left( \sum_{i=1}^{\lceil \frac{n}{f_d} \rceil} m_{i,j} \right)^2 + \sum_{i=1}^{n} \left[ \left( \sum_{j=1}^{f_d} m_{i,j} \right) - 1 \right]^2 \right). \quad (1)$$

**Theorem 3.1.** *In the case of the global optimum with $\mathcal{L}_S = 0$ and under the requirement that $s(p_i) \geq 0, \forall p_i \in P$, there cannot be two neighboring selected points.*

*Proof.* It is sufficient to show that the matrix until row $\lceil \frac{n}{f_d} \rceil$ will cause $\mathcal{L}_S > 0$ as $\sum_{i=\lceil \frac{n}{f_d} \rceil+1}^{n} ((\sum_{j=1}^{f_d} m_{i,j}) - 1)^2 \geq 0$. To this end, we show that in the simplest setting $\mathcal{L}_S > 0$ if two neighboring points are selected. We proceed to show that if a matrix already caused $\mathcal{L}_S > 0$ independent of the particular deviation, neither an addition of a row nor a column can lead to $\mathcal{L}_S = 0$ which completes the induction. For the global optimum of $\mathcal{L}_S = 0$ all row sums $r_i = \sum_{j=1}^{f_d} m_{i,j}$ must equal the optimal value $r_i^* = 1, \forall i$ and all column sums $c_j = \sum_{i=1}^{\lceil \frac{n}{f_d} \rceil} m_{i,j}$ must equal the optimal value $c_1^* = \lceil \frac{n}{f_d} \rceil$ and $c_j^* = 0, \forall j \in \{2, \ldots, f_d\}$. In the simplest case of $f_d = 2$ and two selected points $p_1$ and $p_2$ (i.e., $n \in \{3, 4\}$) with attention values $s(p_1) = a_1 > s(p_2) = a_2$ if $p_1$ is a

---

[2]Note that the value corresponding to a node being selected can be any value $> 0$ if the network does not need too large weights to attain it. Here, 1 is chosen as an analogy to *true* and *false*.

Table 1: Illustration of the ideal setting for diversely selected points if $f_d = 3$. $p_1$ and $p_2$ are the selected points (blue) and receive an importance value of 1 (first value column $s(p_i)$ of the table depicting $M$). Their $f_d$-nearest-neighbor neighborhood is represented by the orange lines. All other points receive an importance value of 0 but either $p_1$ or $p_2$ is their first nearest neighbor ($NN_1$).

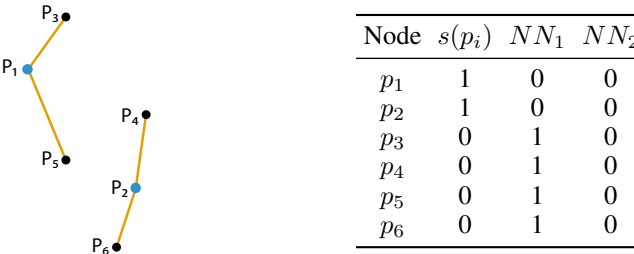

| Node | $s(p_i)$ | $NN_1$ | $NN_2$ |
|------|----------|--------|--------|
| $p_1$ | 1 | 0 | 0 |
| $p_2$ | 1 | 0 | 0 |
| $p_3$ | 0 | 1 | 0 |
| $p_4$ | 0 | 1 | 0 |
| $p_5$ | 0 | 1 | 0 |
| $p_6$ | 0 | 1 | 0 |

nearest neighbor of $p_2$ to attain the global optimum for $\mathcal{L}_S$ it is required that $c_1 = a_1 + a_2 \overset{!}{=} c_1^* = 2$ and $r_2 = a_1 + a_2 \overset{!}{=} r_2^* = 1$ yielding a contradiction. Thus, either $d_{c,1} = c_1 - c_1^* \neq 0$ or $d_{r,2} = r_2 - r_2^* \neq 0$ or both. In the general case $\exists j : d_{c,j} \neq 0 \lor \exists i : d_{r,i} \neq 0$ and thus

$$d = \sum_{i=1}^{\lceil \frac{n}{f_d} \rceil} |d_{r,i}| + \sum_{j=1}^{f_d} |d_{c,j}| > 0. \tag{2}$$

If adding a new column by increasing $f_d$ to $f_d + 1$ could cause $d = 0$ its attention values must be $m_{i,f_d+1} = -d_{r,i}, \forall i$ and $\exists i : d_{r,i} \neq 0 \land \nexists j : d_{c,j} \neq 0$ needs to be true. Thus, the loss requires

$$c_{f_d+1}^* = 0 \overset{!}{=} \sum_{i=1}^{\lceil n/f_d \rceil} -d_{r,i}. \tag{3}$$

Per requirement there can only be $m_{i,j} \geq 0$ and therefore $d_{r,i} \leq 0, \forall i$ which yields a sum greater than zero and thus a contradiction to 3. If adding a new row by increasing the number of selected points could cause $d = 0$, its attention values must be $m_{\lceil \frac{n}{f_d} \rceil+1,1} = -d_{c,1} + 1$ and $m_{\lceil \frac{n}{f_d} \rceil+1,j} = -d_{c,j}, \forall j \in \{2, \ldots, f_d\}$ since $c_1^*$ increases by 1 if a new row is added. Further, analogously to above $\exists j : d_{c,j} \neq 0 \land \nexists i : d_{r,i} \neq 0$ must be true. Hence, for this case $g = 0$ requires

$$r_{\lceil n/f_d \rceil+1}^* = 1 \overset{!}{=} (-d_{c,1} + 1) + \sum_{j=2}^{f_d} -d_{c,j} = \sum_{j=1}^{f_d} -d_{c,j} + 1 \qquad \Rightarrow \qquad 0 \overset{!}{=} \sum_{j=1}^{f_d} -d_{c,j}. \tag{4}$$

The argument now is the same as it was for adding a new column. Thus, adding rows and columns cannot change $d$ to be equal to zero which is the requirement for the global optimum and therefore, two selected points neighboring each other yields $\mathcal{L}_S \neq 0$. $\qquad\square$

While in theory, the proof requires that $s(p_i) \geq 0, \forall p_i \in P$, our experiments have shown that in practice, it is sufficient to employ a LeakyReLU instead of a ReLU activation function on the predictions. Advantageously, the complete selection operation causes the network to learn an order of nodes from the point cloud represented by the attention vector. This order will be independent of the order of nodes in the input tensor as all operations performed are permutation invariant. The independence property of point clouds enables the construction of autoencoders for point clouds that can learn a common representation for it, disregarding the input permutation of points. Further, a reduced point subset allows the network to connect nodes previously far apart from one another.

**Transposed Convolutions** Upsampling of feature maps, in the case of CNNs for images, often happens via transposed convolutions. Transposed convolutions are also referred to as convolutions with fractional strides. This step size, smaller than one, is obtained by adding spacing between the entries of the feature map. Thus, a kernel processes neighboring relations in a less dense but rather more spacious manner. If a stride of $\frac{1}{2}$ is applied the distance between positions in the input is doubled and the void positions are filled with zero entries. On this new feature map, the kernel is applied

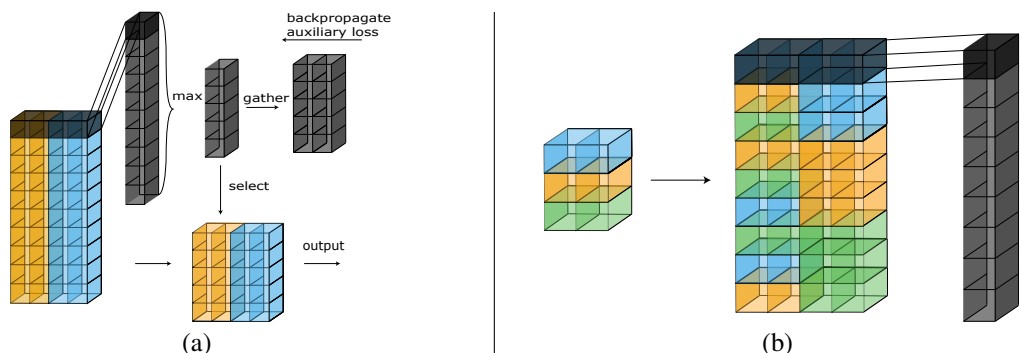

$$(a) \hspace{10em} (b)$$

Figure 1: (a) Illustration of the proposed convolution with a stride greater than one on point clouds. The input tensor (omitting the batch dimension) consists of the points in orange (two channels for two-dimensional points) and the gathered $k$-nearest-neighbor points attached to them in blue (with $k = 1$ for visibility reasons). Potentially, multiple layers of $1 \times 1$ kernel convolutions (grey box over the input tensor) operate on this tensor and from the resulting importance vector, the $\lceil \frac{n}{f_d} \rceil$ values with the highest score are selected (curly braces). The input feature map is reduced to the corresponding nodes (bottom right) and the auxiliary loss is computed based on the importance vector (top right). (b) Illustration of the upsampling operation on point clouds. The exemplary input feature map (left) contains 3 points (shown by different colors). Every $k$-nearest-neighbor relationship is processed individually (middle) leading to an increase of the point dimension.

216 with a step size of one. Thus, in most cases, only two previous positions are covered by a kernel.
217 Compliant with the desired reproduction of convolution operations for point clouds during the novel
218 upsampling operation, the neighboring relationships are considered individually. In this operation, a
219 kernel creates a feature of a new point by processing the current point and one of its nearest neighbors
220 without knowledge of the point positions which should be sampled. This way, learned information
221 about the surrounding of a current point captured in its feature vector can be translated into new
222 points. This operation can be seen in Figure 3. Specifically, every point $p_i$ is repeated $f_u$ times and
223 stacked with the vectors pointing from $p_i$ to its $f_u - 1$ nearest neighbors and the null vector. This
224 yields $n * f_u$ different point vector pairs which are processed by a normal convolution operation
225 producing $n * f_u$ points in a new feature space. This way, the convolutions with a stride greater than
226 one can be undone and higher-level features can be translated into lower-level ones.

## 4 Experiments

228 Autoencoders make use of down- and upsampling operations, and hence, are suitable to exemplarily
229 apply both presented proxies for the point cloud domain. The encoder of the network presented in
230 this work is analogous to the well-known ResNet structure [9], with the additional auxiliary loss
231 $\mathcal{L}_S$ from Equation 1 enabling strided convolutions. The decoder structure also employs the ResNet
232 blocks and implements the concept for transposed convolutions. For a more detailed description of
233 the architecture specifics see the appendix. To test the performance of our proposed selection strategy
234 in other architectures, we replace the FPS module in Point-M2AE with our FPS alternative. However,
235 incorporating our selection into the Point-M2AE Transformer is not straightforward since their
236 masking strategy depends on the selection module in a way that the neighborhood embeddings of the
237 unmasked points do not interfere with those of the masked points. Consequently, the selection must
238 be completed before the network calculations are carried out. Our proposed selection is intentionally
239 based on the previous layer output. This way the selection module can leverage the knowledge of what
240 points cause high activations in previous layers. Incorporating this selection after the computation
241 of the different hierarchical layers, however, enables the network to select points at the boundary
242 between masked and unmasked tokens in the transformer and causes the training to not converge.
243 Therefore, we build two versions: Point-M2AE-c with a selection module only processing the spatial
244 information of the points and Point-M2AE-e for which the first selection module is based on the
245 neighborhood embeddings. These embeddings are independent of the masking and the following
246 selections are again based on the points themselves.

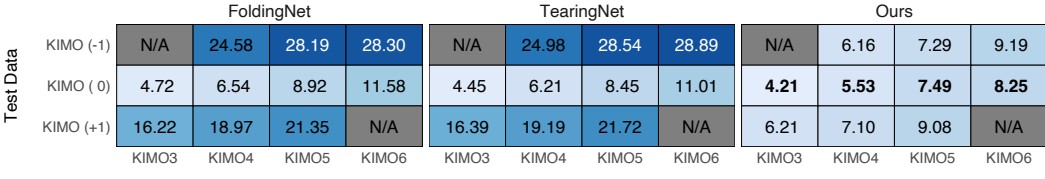

Figure 2: The performance values of the different models given in terms of the extended chamfer distance (Equation 5), multiplied by factor 100 for better readability. Each of the models (represented by three separate blocks) was trained on four training data sets (columns per block) and tested with those validations sets deviating in their grid size by $\pm 1$ (rows). The best performing not cross tested model is marked in bold.

## 4.1 Point Cloud Reconstruction

In the first experiment, we investigate on the reconstruction ability of the proposed autoencoder and train it as well as Folding- and TearingNet[3] on the KIMO3-6 data sets proposed by [13], synthesized out of the KITTI 3D object data set [7] by cutting out traffic participants, and designed with challenging multiple-object scenes in mind. In our experiments, each data set contains $50,000$ training instances and $10,000$ test instances, deviating from the original construction description of the data sets to make the trained models more comparable. Therefore, Tearing- and FoldingNet are trained with their default parameters. Our own networks are trained with a learning rate of $4 \cdot 10^{-3}$ which is exponentially decreasing and halved every 80 epochs. The $k$ is set to 10 and the $f_d$ and $f_u$ values are set to 6 for every layer, yielding a code word that consists of $57$ points described each by $9$ code word channels (cc). The performance of all models is evaluated with the extended chamfer distance [26, 13] between the reconstructed and the original point cloud. This metric between two point sets $S_1$ and $S_2$ is defined as

$$d(S_1, S_2) = \max\left\{\frac{1}{|S_1|}\sum_{x_1 \in S_1}\min_{x_2 \in S_2}\|x_1 - x_2\|^2, \quad \frac{1}{|S_2|}\sum_{x_2 \in S_2}\min_{x_1 \in S_1}\|x_2 - x_1\|^2\right\}. \tag{5}$$

The chamfer distance is the loss function of the Folding- and TearingNet. For our network, we found a slightly improved performance when using squared distances between points in the loss function. We analyze the reconstruction capabilities of the different networks (a) for the same type of data set that was used for training, and (b) across corresponding neighboring data sets (see Figure 2). To ensure better comparability of the results for the cross-data set studies, we scaled all point clouds to the size of the point clouds that were used for training the respective model. The results can be seen in Figure 2. Our approach outperforms the state-of-the-art models, when the test data corresponds to the training data. The most significant performance gain is achieved for the KIMO6 data set with many small objects on average. Notably, our proposed approach is very robust, as evidenced by the competitive performance on the neighboring data sets. This means that it can handle other data sets much better (i.e., it generalizes more) than its two main competitors. However, in our approach, the typically superior model is the one trained on the corresponding training data. There is an anomaly with KIMO5, as the models trained on KIMO4 outperform it. In contrast, all other models trained on non-corresponding training data achieved slightly inferior, yet still comparable, results.

## 4.2 Classification

To assess the capability of our autoencoder to effectively represent 3D data in the generated code words, following the methodology employed in prior studies (see Table 2 on the left), we conduct a two-step evaluation. First, we train the autoencoder on the ShapeNet [4] data set, which encompasses

---

[3]Note that for the task of compressing the complete information of the scene and reconstructing the original point cloud, we are not directly comparable with existing transformers like Point-M2AE. This is because they are trained using a masking strategy to learn as much global information as possible and do not compress information as networks trained to fully reconstruct a point cloud do. Both approaches are beneficial in different use cases.

Table 2: The achieved accuracy of an SVM trained on the representation obtained by different self-supervised learning models for the ModelNet40 data set.

| Model | Acc. (%) |
|---|---|
| T-L Network [8] | 65.4 |
| 3D-GAN [22] | 83.3 |
| Latent-GAN [1] | 84.5 |
| FoldingNet [26] | 88.4 |
| DGCNN + CrossPoint [2] | 91.2 |
| Transformer + OcCo [27] | 89.6 |
| Point-BERT [27] | 87.4 |
| Point-M2AE [28] | **92.9** |

| Model | Subsampling | $f_d$ | Acc. (%) |
|---|---|---|---|
| Config-1 | ours | (6, 6) | 37.1 |
| Config-2 | ours | (14, 8) | 67.0 |
| Config-3 | ours | (12, 12) | 81.8 |
| Point-M2AE-fps | fps | (8, 4, 2) | 91.6 |
| Point-M2AE-c | ours | (4, 2, 4) | 90.3 |
| Point-M2AE-e-1 | ours + emb | (4, 2, 4) | 91.2 |
| Point-M2AE-e-2 | ours + emb | (8, 4, 2) | 90.7 |

55 distinct 3D object categories and over 50,000 3D shapes. Then, we store the code words of the previously unseen 3D objects in the ModelNet40 [23] data set and aggregate the information by taking the sum of the mean and maximum for every code word. Utilizing those aggregated code words, we train a linear Support Vector Machine (SVM) as our classifier. We conduct experiments involving different variations of our architecture, as well as variants of Point-M2AE, by replacing FPS with our own selection approach. The results of these experiments are presented in Table 2.

We test our network in three different configurations to vary the amount of features and points in the code word. The first one, "Config-1", is equivalent to the network configuration used during point cloud reconstruction. Despite outperforming its competitors during the reconstruction task, the SVM struggles to effectively distinguish the code word representations. We hypothesize that this can be partially attributed to the limited nature of our network's code word, which comprises 57 points with 9 cc each. By contrast, the competing model with the highest accuracy "Point-M2AE" has 64 points with 348 cc. If we alter the composition of our model's code word to 19 points and 27 cc ("Config-2") we are able to increase the SVM accuracy by almost $50\%$ without changing the total information of the code word ($57 \times 9 = 19 \times 27$). If we allow more information in the code word with "Config-3" (15 points and 137 cc) we are able to achieve a competitive accuracy while still maintaining significantly less information in the code word ($15 \times 137 < 64 \times 348$). In addition to the aforementioned comparisons, it is crucial to consider the parameter count of the models being compared. For instance, Point-M2AE utilizes approximately $15.3 \times 10^6$ parameters, while FoldingNet employs around $2 \times 10^6$ parameters. In contrast, our network operates with a substantially lower parameter count of approximately $3.4 \times 10^5$. This disparity in parameter count highlights an important aspect to consider when evaluating the efficiency and computational requirements of the different models under investigation.

As expected, Point-M2AE-c with a selection only based on the spatial performance of the points performs worse than Point-M2AE-e-1, which, in turn, however, cannot achieve the same performance as the original Point-M2AE with the equivalent $f_d$ configurations. Nevertheless, the new selection strategy proves to be more robust when a large $f_d$ is employed. It can be seen that the decrease in performance is not as significant for Point-M2AE-e-2 compared to Point-M2AE-e-1 as for Point-M2AE-fps compared to Point-M2AE.

### 4.3 Ablation

In the ablation study, we investigate the influence of the choice for the number of nearest neighbors per point processed within one convolution and the choice of the $f_d$ and $f_u$ values. The reconstruction quality does not differ significantly for all tested parameters and stabilizes by a chamfer distance of approximately $0.018$, suggesting that $f_d$ and $f_u$ are not the bottleneck for the compression of information – encouraging a lighter parameter choice leading to computationally less expensive models. Further, we implement FPS in our proposed architecture instead of a network guided sampling and find that we can achieve the same reconstruction performance as FPS in our architecture while producing a permutation invariant code word. The detailed results can be found in the appendix.

To get a better understanding of the model's learning process, the distribution of the selected points can be analyzed, e.g., using the point sampling depicted in Figure 3. The blue points are selected in the first level and the red points in both the first and the second level. If the points are not distributed

| Epoch 1 | Epoch 50 | Epoch 300 |
|---|---|---|

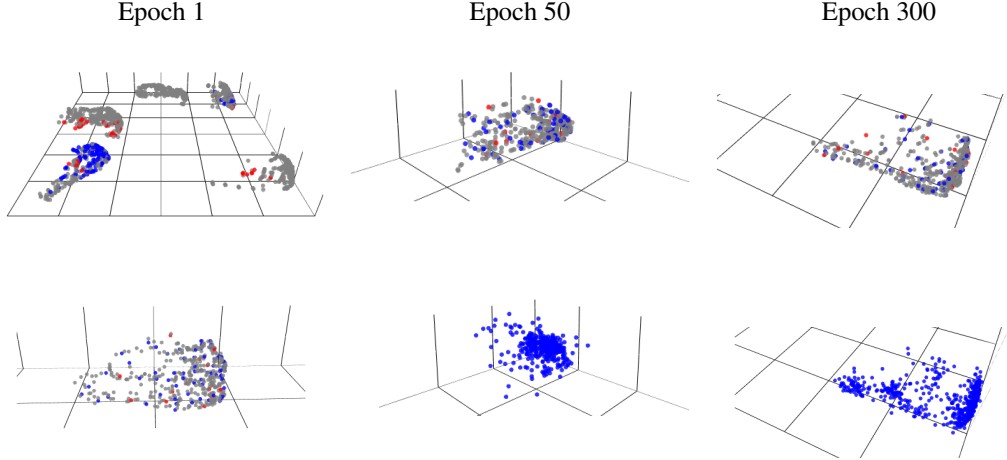

Figure 3: The model's point selection (top row and bottom left), and model predictions (bottom middle and bottom right). Columns correspond to the different epochs (1, 50, and 300) and depict different aspects of the learning procedure.

well the network will (during the upsampling process) tend to clump up points in the dense areas and produce sparse regions for areas that have only a few points selected. This process is controlled by the auxiliary loss $\mathcal{L}_S$, and its influence can be observed in the left column of Figure 3. The model demonstrates inadequate point distribution during the start of the first epoch (top left), whereas, by the end of the first epoch (bottom left), it has learned to distribute the points. However, simply distributing the points evenly will result in fuzzy edges and poor capturing of surfaces, as can be seen in the predictions in the middle column of Figure 3, which displays the original object (top middle) and its prediction (bottom middle), respectively, after roughly 50 epochs of training. This prediction shows that the model has learned to locate the object and pinpoint the center of mass, but has not captured any exact surfaces yet. Also, there are some points that are distributed completely outside the desired shape. This happens because the model has problems bounding the figure during the upsampling process. Thus, it is also important for the network to specifically select points that are at the corner or edges of an object. In the top right of Figure 3, it can be seen that, specifically, the red points are very frequently distributed at the contour in a model that has been trained for roughly 300 epochs. On the bottom right, the corresponding prediction is depicted. The quality of the prediction can be visually assessed, as the points form even surfaces with only minimal deviations. Furthermore, there are no outliers that scatter far away from their intended position.

## 5    Conclusion

In conclusion, we introduce a novel network-based point selection strategy that guarantees the diversity of selected points equivalent to FPS, while possessing the advantages of permutation invariance and learnability. Employing the proposed strategy in a simple yet effective autoencoder we show its superiority compared to previous state-of-the-art approaches on the task of fully reconstructing a point cloud with multiple objects. Interestingly, both established methods had difficulties processing data types on which they were not trained, while our proposed model generalizes much better and suffers only minor performance losses. Even though the model is considerably smaller than recent unsupervised learning models it is able to represent a given 3D shape well. Further, the proposed FPS alternative proves to be integrateable into other existing architectures and is more helpful for the model if based on hierarchical features learned by the model rather than the spatial locations of the points. For future work, we plan to integrate our procedure into a transformer architecture trained with a masking strategy not dependent on the selected points and further investigate on more complex architectures. Moreover, synergies between our selection strategy focusing on diversity, and strategies targeting other point subsets, e.g., ones with little noise, are promising to investigate with a combined auxiliary loss function.

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
