# OpenReview forum: "Deterministic Strided and Transposed Convolutions for Point Clouds Operating Directly on the Points"
_NeurIPS.cc/2023/Conference — Submitted to NeurIPS 2023_

### Official Review · Reviewer_KAHP · 2023-07-06

**Soundness:** 2 fair
**Presentation:** 1 poor
**Contribution:** 2 fair
**Rating:** 3
**Confidence:** 4

**Summary:**

This paper focuses on applying strided and transposed convolutions to point cloud data so that the deterministic network can be directly operated on points. To achieve this, a strided convolutional layer with auxiliary loss is proposed, which ensures a consistent selection of points across the whole learning process. Further, a lightweight autoencoder network is built upon the proposed convolutional operator. Experiments are conducted on KIMO3-6 for point cloud reconstruction and ModelNet40 for shape classification.

**Strengths:**

1.	The proposed idea and method of performing strided and transposed convolutions on point clouds are interesting.
2.	The related work provides an exhaustive discussion of existing point cloud CNNs, which is meaningful.
3.	Fig. 3 intuitively illustrates the point selection during the learning procedure.
4.	Code implementations are provided in the supplementary material.

**Weaknesses:**

**Main weaknesses**:
1.	Poor writing quality:
A.	The whole paper is presented with very long text without clear paragraph/subsection division, which significantly hurts the reading of the paper.
B.	Line 228-246 state the network design for applying the proposed convolution operator, but no figure illustration is provided, making it hard to fully understand how to apply this on Point-M2AE.
2.	This paper is highly related to SparseConvNet [1] which also uses strided convolution on point clouds. However, it is not included, discussed, or compared. This is very important to evaluate the novelty of the proposed method.
3.	Use/comparison of Point-M2AE:
A.	It is not clear why the proposed method is applied on Point-M2AE. Is the method only suitable for auto-encoder-style networks?
B.	In Point-M2AE, in addition to SVM evaluation on ModelNet40, the general/few-shot classification on ModelNet40 and ScanObjectNN, part segmentation on ShapeNetPart, and 3D object detection on ScanNetV2. To fully verify the effectiveness of the proposed method, more experiments should be conducted under some of the benchmark settings as in Point-M2AE.
4.	The reported results are not promising compared with SOTA point CNNs, as shown in Table 2, although fewer #Params are introduced. As a result, this work does not clearly prove the potential of using the proposed 2D-like strided and transposed convolutions, instead of the existing customized 3D point cloud convolutions.
5.	Line 298 states the #params to show the lightweight property of the proposed method. However, the FLOPs and latency are also important to evaluate the efficiency, thus more comparisons of FLOPs and latency should be provided.
6.	The model's robustness to permutation, translation, rotation, scaling, and noise is not tested through experiments, which is important for real-world applications.

**Refs**:
[1] 3D Semantic Segmentation with Submanifold Sparse Convolutional Networks, CVPR 2018.
[2] A closer look at local aggregation operators in point cloud analysis. ECCV, 2020.

**Additional comment**:
PosPool [2] is another representative point convolution method, which is also directly applied to ResNet architecture. It should be included, discussed and compared.

**Questions:**

The idea and the proposed method to perform strided and transposed convolutions on point clouds are interesting. However, a very related work, SparseConv is not discussed and compared, which is very important for evaluating the novelty of the proposed method. Some discussions (better with figures or tables) are expected to be provided.

Moreover, the presentation quality (e.g., poor organization, the lack of figure illustration of network designs) of the whole paper is not qualified for the top-tier conference, which significantly hurts the reading of the paper. It shows that the paper is finished in a rush.
Additionally, some important experiments are not provided, which is important for verifying the effectiveness of the proposed method.

To conclude, the current version is somehow below the bar for a main conference paper, and solving the mentioned weaknesses is non-trivial. Thus, accepting this paper may not be fair considering the deadline is the same for all papers. It is highly recommended to revise the paper carefully, **identify the major bottleneck of performance and try to improve it**, and submit it to another conference.

---

> ### Author Rebuttal · Authors · 2023-08-09
>
> Dear Reviewer,
>
> First and foremost, thank you very much for taking the time to review our paper and providing us with helpful comments to improve it. We are looking forward to discussing the contents with you.
>
> We agree that more figures could help the understanding of the paper and will add them in the future.
>
> Nevertheless, we cannot directly see why you consider SparseConvNet to be highly relevant to our work. In our understanding, SparseConvNet uses a clever trick to keep the higher feature levels sparse and thus performs computations with higher efficiency. However, it also operates on a grid structure for point clouds and is therefore very different from our approach which works directly on the points. Could you tell us where we misunderstood or elaborate more on why you think it is crucial to include this reference? We think that it is very interesting work and will add it to the related work section, but do not see the need for an in-depth discussion.
>
> We applied our method to M2AE as it is the state of the art for unsupervised learning on point clouds. However, you are right that more experiments with different architectures would improve the paper. Could we convince you as a reviewer of the importance of our sampling method if we showed in further experiments that we can outperform FPS in certain settings, e.g. certain transformer models?
>
> Best regards and thank you again for your time
>
> Authors

---

> > ### Comment · Reviewer_KAHP · 2023-08-16
> >
> > This rebuttal does not fully address my concerns. This rebuttal seems more like an argument letter, without any sufficient results or figures. I highly recommend authors read some rebuttals available at the open-review website, such as the rebuttal of the ICLR conference.
> >
> > More importantly, only one backbone, PointM2AE is selected to apply the proposed method, and the experimental settings are not complete (in PointMAE, they did the general/few-shot classification on ModelNet40 and ScanObjectNN, part segmentation on ShapeNetPart, and 3D object detection on ScanNetV2, which are missed in this paper). Moreover, the experimental results are not impressive. At the experimental level, this method is for improving the performance of point cloud backbones under full supervision. It is hard to accept such a quality of experiments.
> >
> > As also indicated by other reviewers, the writing quality of this paper is not acceptable for a top-tier conference and needs non-trivial effort to revise the paper.
> >
> > Therefore, I will certainly keep my original rating of 3.

---

### Official Review · Reviewer_eM7t · 2023-07-07

**Soundness:** 2 fair
**Presentation:** 1 poor
**Contribution:** 2 fair
**Rating:** 3
**Confidence:** 4

**Summary:**

This paper introduces a learning-based point sampling strategy to deterministically downsample point clouds, which can be used to build a U-shaped network for point cloud reconstruction and representation. To enforce a stable and meaningful sampling (or selection), an auxiliary selection loss is proposed. The auxiliary selection loss enables a network to select central points which are likely to be non-neighboring each other. With the sampling strategy, this paper finally proposes a deterministic strided and transposed convolution for point clouds. The proposed method is evaluated in point cloud reconstruction and representation learning (especially SVM classification on the representation) tasks. In the point cloud reconstruction task, the proposed method shows a lower chamfer distance than the previous methods. The ModelNet 40 experiments show that the proposed method can be integrated with various network configurations.

**Strengths:**

1. [Originality] The proposed sampling strategy and its application to the strided convolutions are interesting. Since many previous point cloud networks utilize farthest point sampling as described in the paper (L62-83), the proposed sampling strategy could be a new alternative and bring robustness to those networks.
2. [Clarity] This paper provides detailed explanations of the auxiliary selection loss with a theorem, its proof, and an example. Especially the example (Table 1) explicitly shows how the attention map matrix (M) is constructed and how the selected points are non-neighboring each other.

**Weaknesses:**

1. [Originality] An highly relevant reference, SampleNet [1], is missing in both related work and experiment sections. Since both SampleNet [1] and this paper propose a learning-based point sampling, I recommend the authors explain how the proposed method differs from the SampleNet and evaluation results on the same experiments SampleNet did; supervised classification on ModelNet40.
2. [Quality] The writing quality and layout of the paper should be improved. For example, the proof of Theorem 3.1 and its example (Table 1) can be moved to the Appendix, although they may help readers to understand what the auxiliary selection loss is. Instead of the proof and example, detailed experiment results (e.g., downstream task results) with the proposed method would be better to be added.
3. [Significance] The current setup of experiments is not enough to show the significance of the proposed sampling strategy. Since various networks with farthest point sampling can use the proposed sampling strategy as an alternative, the downstream task results, which those networks did, should be added. For example, Point Transformers [2, 3] with the proposed sampling strategy can be evaluated in 3D semantic segmentation task on S3DIS or shape classification on ModelNet40. I recommend the authors evaluate the proposed sampling strategy with SOTA networks [2, 3] on downstream tasks (e.g., shape classification, semantic segmentation, and registration).

[1] Lang et al., “SampleNet: Differentiable Point Cloud Sampling”, CVPR, 2020.\
[2] Zhao et al., “Point Transformer”, ICCV, 2021\
[3] Wu et al., “Point Transformer V2: Grouped Vector Attention and Partition-based Pooling”, NeurIPS, 2022.

**Questions:**

As described in the weakness section, I have several questions about the contribution of the proposed method and experiments (please see the weakness section for details):

1-1. Compared to SampleNet [1], what are the strengths of the proposed method? \
1-2. Can those strengths be quantitatively evaluated on the downstream task (e.g., shape classification) SampleNet did? \
2. Does the proposed sampling strategy outperform farthest point sampling or voxel subsampling on downstream tasks (shape classification, semantic segmentation, and registration)?

[1] Lang et al., “SampleNet: Differentiable Point Cloud Sampling”, CVPR, 2020.

**Limitations:**

The authors partially addressed their work's limitations in the experiment section but did not address the potential negative societal impact. However, I don't think that there is a particular negative societal impact of this work.

---

> ### Author Rebuttal · Authors · 2023-08-09
>
> Dear Reviewer,
>
> First and foremost, thank you very much for taking the time to review our paper and providing us with helpful comments to improve it. We are looking forward to discussing the contents with you.
>
> Thank you for pointing out SampleNet. We agree that it is highly relevant to our work and will add it to the related work section. Nevertheless, we think that it is different from our approach for two reasons: First, it does not enforce diversity, and second, it is trained with an iterative training procedure. This means that first, the actual network is trained on the task, then its weights are fixed and SampleNet is trained so that the selection improves the task performance. This procedure creates difficulties when multiple hierarchy levels are desired.
>
> You further mention that the writing quality and layout should be improved. We agree that the layout would benefit from moving the example to the appendix in favor of showing more experiments. However, we were not sure which parts of the paper were of poor writing quality. Could you point us to specific sections?
>
> Replacing farthest point sampling in other architectures is a helpful suggestion and we will do this in future experiments. Could we convince you as a reviewer of the importance of our sampling method if we showed in further experiments that we can outperform FPS in certain settings, e.g. certain transformer models?
>
> Best regards and thank you again for your time
>
> Authors

---

> > ### Comment · Reviewer_eM7t · 2023-08-22
> > **Official Comment by Reviewer eM7t**
> >
> > Thank you for the rebuttal. I have read the rebuttal and found that it did not address my initial concerns (quantitative comparison with SampleNet [Lang et al., 2020] and farthest point sampling). Therefore, I will keep my initial rating (reject).

---

### Official Review · Reviewer_MuCB · 2023-07-09

**Soundness:** 2 fair
**Presentation:** 1 poor
**Contribution:** 1 poor
**Rating:** 3
**Confidence:** 4

**Summary:**

This paper presents a learnable and deterministic point selection layer to uniformly downsample points and a point transposed convolution layer to upsample points.

**Strengths:**

1. The auxiliary loss (Eqn. 1) proposed to supervise the point selection is interesting.

**Weaknesses:**

1. Deficient theoretical soundness. The proposed downsampling layer attempts to learn a point importance score and selects the points with the highest scores. Given that the selection operation is non-differentiable, your importance prediction network is solely supervised by the auxiliary loss (which functions as a uniform sampling regularization). As a result, it appears to be optimized to output uniformly sampled points rather than points based on semantic importance. However, the authors seem to disagree and claim that their proposed subsampling layer is capable of learning how to select points based on semantic importance (L330). Please elucidate how this non-differentiable operation can learn a importance-based sampling. It is worth noting that previous work leverages Gumbel-Softmax to enable a soft learnable selection, which results in a differentiable point subsampling network (L105).

2. Implementation and trade-off details of the learnable subsampling? Is it implemented using a single linear layer as depicted in Figure 1? How efficient it is? How many additional parameters are required? How practical is it to scale this up to a large-scale point cloud?

3. Missing large-scale experiments. What is the performance of the classical PointNet++ or the latest PointNeXt with the proposed subsampling in a large-scale dataset like ScanNet?

4. Visualizations appear to be missing. It would be insightful to see how the selected points differ from FPS.

5. No improvement over FPS (Table 2).

**Questions:**

Figure 3 is not clear. I do not understand why the samples change with epoch. Why not showing how the point subsampling changes with the learning epochs using the same sample.

**Limitations:**

In which specific applications is deterministic downsampling critical? Based on the results presented in Table 2, I observed no performance improvement, but rather a decline, when using the proposed downsampling method compared to FPS.

For many applications, the determinism isn't a concern as the network is resilient to minor variances in subsampled points. This is particularly true in the case of large-scale point clouds, where the variance in subsampled points is typically small. Could you clarify the necessity and advantages of your deterministic downsampling method in this context?

---

> ### Author Rebuttal · Authors · 2023-08-09
>
> Dear Reviewer,
>
> First and foremost, thank you very much for taking the time to review our paper and providing us with helpful comments to improve it. We are looking forward to discussing the contents with you.
>
> Regarding the first weakness mentioning the discrepancy between the selection being based only on the auxiliary loss and the observation of a semantic selection, we fully agree with you that the selection is not differentiable with respect to the task loss. We did not mean to say otherwise.
>
> However, during the analysis of the selection procedure we observed that the selected points tend to be those with a high activation for the task-dependent features. Thus, we hypothesized that the selection of non-neighboring points guided by the auxiliary loss is easier for the network if it takes semantic information into account. That is why we think that the selection is influenced by semantic information. Would you need more proof for this hypothesis?
>
> Our subsampling is implemented using two bottleneck ResNet blocks, and the number of parameters depends on the number of channels used. In the case of 64 channels in the previous block, there are 50707 additional parameters. The module does not scale well with the increasing size of the point cloud due to the nearest neighbor computations, but this is also true for FPS.
>
> Figure 3 showed different samples because they revealed different properties of the selection, but we agree that finding a sample that shows all properties would help the reader to better comprehend the figure. We will work on enhanced visualizations.
>
> The permutation invariance property is desirable as it increases the guaranteed robustness of the output, however, we agree with you that we should directly show this benefit in further experiments.
>
> Could we convince you as a reviewer of the importance of our sampling method if we showed in further experiments that we can outperform FPS in certain settings, e.g. certain transformer models?
>
> Best regards and thank you again for your time
>
> Authors

---

### Official Review · Reviewer_Dxvh · 2023-07-11

**Soundness:** 2 fair
**Presentation:** 2 fair
**Contribution:** 1 poor
**Rating:** 4
**Confidence:** 4

**Summary:**

This paper aims to propose a new type of convolutional neural network to point cloud understanding tasks. Besides, the authors present a new loss function to the sampling process of feature extraction for point clouds. And the authors provide theoretical analysis to prove that their method is better for sampling points. The authors conduct experiments on the reconstruction and shape classification tasks. The experimental results on shape classification are not satisfactory.

**Strengths:**

- Theoretical analysis for the proposed algorithm.
- Better performance on the reconstruction tasks.

**Weaknesses:**

- Poor writing.
- Unclear motivation.
- Unsatisfactory experimental results for point cloud understanding.
- Inaccurate statements.

**Questions:**

- In L3-4, the authors wrote that "point clouds are less structured than images". I guess that it should be "Point clouds contain more complicated structural information than images."
- Why do we need to develop convolution operators for point cloud understanding? For example, with simple geometric projection, we can transform the original point cloud into image grids. It's more efficient and reasonable. P2P [a] with a tiny-scale ConvNeXt can outperform Point-M2AE-e-1 proposed in this paper.
- Can we replace the FPS operation with the proposed method? From a practical view, existing FPS operators have been enhanced with the CUDA library, which brings higher computation efficiency. Can the authors provide additional information about the difference in the execution time between the proposed method and the FPS operator?
- I can not directly get the advantages of the proposed method, compared with existing methods. Can the authors summarize the advantages of training parameters, inference speed, and experimental performance?

**Limitations:**

Lack of limitation discussion.

---

> ### Author Rebuttal · Authors · 2023-08-09
>
> Dear Reviewer,
>
> First and foremost, thank you very much for taking the time to review our paper and providing us with helpful comments to improve it. We are looking forward to discussing the contents with you.
>
> Among the critical points you raised, you noted deficiencies in the writing of our paper and that we made inaccurate statements. Could you please elaborate on what parts were not well written? You mention that we wrote “point clouds are less structured than images” in the abstract. We meant it in the sense of organization of the points. An image can be thought of as a point cloud where the points have color information attached and are organized in a grid. A point cloud in general does not fulfil this property. Would you agree with this explanation? Are there further parts in the text that you find misleading?
>
> Moreover, thank you for the pointer to the paper “P2P: Tuning Pretrained Image Models for Point Cloud Analysis with Point-to-Pixel Prompting”. The approach is very interesting. However, it relies on a pre-trained image transformer and first utilizes DGCNN for point cloud understanding. Thus, in our opinion, this does not show that there is no need for research on convolution operators for point cloud understanding, as with increased datasets more sophisticated point cloud architectures may again outperform the pre-trained vision transformer potentially utilizing our proposed FPS alternative. Also, the DGCNN model may be improved with our proposed method.
>
> You asked about the execution time of our proposed method. We would like to point out that we have not been able to optimize our code in the same way as FPS, as this is beyond the scope of our current capabilities. Nevertheless, during training we did not encounter severe differences with regards to computation time. Would you consider the paper acceptable only if we could obtain a speed benefit in addition to the permutation invariance?
>
> The overall advantage of our approach is the permutation invariance of the operation. In the experiments we could show that our relatively small model outperforms TearingNet on the complete reconstruction task. However, we agree that further experiments demonstrating this advantage on different tasks are desirable.
>
> Could we convince you as a reviewer of the importance of our sampling method if we showed in further experiments that we can outperform FPS in certain settings, e.g. certain transformer models?
>
> Best regards and thank you again for your time
>
> Authors

---

> > ### Comment · Reviewer_Dxvh · 2023-08-14
> >
> > I thank the authors for providing a rebuttal response. After carefully reading reviews from other reviewers and responses from the authors. I believe that this paper is potentially meaningful for the area. However, based on this manuscript, I still think that it does not satisfy the standard for publication. Please refer to the points shared below:
> >
> > * "point clouds are less structured than images"
> >
> > I have understood what the authors mean. I think that the authors should also take RGB information into consideration. In addition, I recommend a relevant paper [Image as Set of Points, ICLR 2023] for the authors, which may be helpful for your research.
> >
> > * Convolution Operation for Point Cloud.
> >
> > I agree with the authors that P2P still requires convolution operation. But purely convolution networks like ResNet-50 can also work with P2P for point cloud understanding.
> >
> > * Additional Experimental Results
> >
> > Compared with acceptance, I think that more experimental benefits with the proposed method should be a necessary part of the presentation. This paper is not in good shape, therefore, I lean to reject the paper and I will keep my rating.

---

### Decision · Program_Chairs · 2023-09-21

**Decision:**

Reject

**Comment:**

This paper proposed a new convolutional neural network for point clouds with a learnable and deterministic point selection layer to uniformly downsample points and a point transposed convolution layer to upsample points. Besides, the authors present a new loss function to the sampling process of feature extraction for point clouds. The authors provided theoretical analysis to prove that their method is better for sampling points.

All three reviewers raised concerns about the writing quality and the lack of certain experiments (replacement of FPS, missing larger datasets, no SOTA performance, etc.). The authors provided some feedback but only promise to add new experimental results in the future. It's unsure whether the authors will conduct new experiments.